

# Sulfate reduction and anaerobic oxidation of methane in sediments of the South-Western Barents Sea

Claudio Argentino[1], Kate Alyse Waghorn[1], Stefan Bünz[1], Giuliana Panieri[1]

[1]CAGE - Centre for Arctic Gas Hydrate, Environment and Climate, Department of Geosciences, UiT The Arctic University of Norway, 9037 Tromsø, Norway.

*Correspondence to:* Claudio Argentino (claudio.argentino@uit.no)

**Abstract.** Anaerobic oxidation of methane (AOM) in marine sediments strongly limits the amount of gas reaching the water column and the atmosphere but its efficiency in counteracting future methane emissions at continental margins remains unclear. Small shifts in methane fluxes due to gas hydrate and submarine permafrost destabilization or enhanced methanogenesis in warming Arctic continental shelves may cause the redox boundary in which AOM occurs, known as Sulfate-Methane Transition Zone (SMTZ), to move closer to seafloor, with potential gas release to bottom waters. Here, we investigated the geochemical composition of pore water ($SO_4^{2-}$ and DIC concentration, $\delta^{13}C_{DIC}$) and gas ($CH_4$, $\delta^{13}C_{CH4}$) in eight gravity cores collected from Ingøydjupet trough, South-Western Barents Sea. Our results show a remarkable variability in SMTZ depth, ranging from 3.5 m to 29.2 m, and that all methane is efficiently consumed by AOM within the sediment. From linear fitting of the sulfate concentration profiles, we calculated diffusive sulfate fluxes ranging from 1.5 nmol cm$^{-2}$ d$^{-1}$ to 12.0 nmol cm$^{-2}$ d$^{-1}$. AOM rates obtained for two cores using mixing models are 6.5 nmol cm$^{-2}$ d$^{-1}$ and 6.7 nmol cm$^{-2}$ d$^{-1}$ and account for only 64% and 56% of total sulfate reduction at the SMTZ ($SRR_{tot}$), respectively. The remaining 36% and 44% $SRR_{tot}$ correspond to organoclastic sulfate reduction with rates of 3.7 nmol cm$^{-2}$ d$^{-1}$ and 5.3 nmol cm$^{-2}$ d$^{-1}$. The shallowest SMTZs ($< 5$ m) and largest $SRR_{tot}$ rates are associated with a shallow subsurface accumulation of gas visible in seismic data, highlighting how small changes in sulfate reduction rates linked to subsurface methane gradients resulted in vertical shifts in SMTZ position of $> 20$ m. This study provides new insights into the dynamic and biogeochemistry of the SMTZ in marine sediments of continental margins and may help predict the response of the microbial methane filter to future increase in methane fluxes due to ocean warming.

## 1 Introduction

Despite covering the ~70% of planet's surface, oceans only contribute a small fraction of present-day methane emissions into the atmosphere (3-6%) because most of the gas generated in sedimentary basins becomes oxidized during its upward migration to the seafloor before reaching the water column and atmosphere (James et al., 2016; Weber et al., 2019). Marine sediments host an efficient biological filter for methane, constituted by a microbial consortia that uses sulfate to perform the anaerobic oxidation of methane (AOM) (Boetius et al., 2000). AOM occurs within the Sulfate-Methane Transition Zone (SMTZ), a redox boundary marking the transition between sulfate-rich pore waters and the underlying methanogenic zone. The SMTZ is highly dynamic and rapidly responds ($10^2$-$10^3$ yr) to perturbations in pore water composition due to gas hydrate and permafrost decomposition (Hong et al., 2016; Overduin et al., 2015; Sultan et al., 2016) or emplacement of mass-transport deposits (Hensen et al., 2003; Kasten et al., 2003). At a global scale, sulfate-driven AOM consumes a total of ~45-61 Tg $CH_4$ yr$^{-1}$, which approximately balances the cumulative $CH_4$ production by methanogenesis in the sediments and corresponds to mineralization of ~ 3-4% of the total organic carbon sinking to the seafloor (Egger et al., 2018). Methane fluxes control the depth of the SMTZ (Borowski et al., 1996, 1999; Jorgensen and Kasten, 2006; Kasten et al., 2003), and under steady-state conditions and in non-hydrate bearing sediments they reflect the rates of methanogenesis (Henrichs and Reeburgh, 1987). Climate change is altering the oceanic carbon cycle by favoring





accumulation and preservation of organic matter in coastal sediments (Diaz and Rosenberg, 2008; Middelburg and Levin, 2009; Wallenius et al., 2021) due to the expansion of oxygen minimum zones (Li et al., 2020; Rabalais et al., 2010). Changes in organic matter burial can increase methanogenic production (Egger et al., 2016, 2018), affecting SMTZ stability. Global warming is also

responsible for the observed Atlantification of the Arctic Ocean, increasing the influx of warm water masses in eastern Barents Sea (Lind et al., 2018; Smedsrud et al., 2013). The warming trend involves the entire water column (Lind et al., 2018) and a rise of near-bottom water temperatures has been recently reported (Skagseth et al., 2020). As the activity of methanogens in marine sediments is strongly dependent on in-situ temperature (Maltby et al., 2018), a warming Arctic may result in enhanced methane production within the shallow sedimentary column and eventually lead to a shoaling of the SMTZ in vast areas of the continental

shelf. Ocean warming-related destabilization of gas hydrates and marine permafrost would have the same overall effect on SMTZ dynamic (Borges et al., 2016; James et al., 2016; Stranne et al., 2019). All these processes would lead to a shallow and "compressed" methane oxidation barrier, increasing the net methane export to the ocean, amplifying ocean acidification (Biastoch et al., 2011).

The SW Barents Sea hosts several areas marked by natural seafloor methane emissions and/or associated with subsurface

gas accumulations related to gas hydrates or hydrocarbon leakage from deeper Mesozoic reservoirs (Chand et al., 2009; Vadakkepuliyambatta et al., 2015). This setting offers a unique opportunity to investigate SMTZ dynamic and biogeochemical processes in key areas characterized by spatially variable subsurface methane fluxes, providing insights into future scenarios of climate-driven increasing methane fluxes in coastal environments and Arctic continental shelves affected by ocean warming.

In this paper we investigate the geochemistry of pore water ($SO_4^{2-}$ and DIC concentration, $\delta^{13}C_{DIC}$) and gas in gravity

cores collected from Ingøydjupet trough, SW Barents Sea. This area was selected as it hosts a subsurface seismic anomaly related to gas accumulation and can be used to explore spatial variations in SMTZ depth, sulfate reduction rates and AOM. We used a two-component mixing model based on dissolved inorganic carbon systematics (DIC, $\delta^{13}C_{DIC}$) and constrained by gas isotope analysis ($\delta^{13}C_{CH4}$) to determine the contribution of organic matter mineralization (OSR) to the overall sulfate reduction in the SMTZ and obtain accurate AOM rate values. This study integrates global datasets of diffusive sulfate and methane fluxes in

continental shelves improving the knowledge on the dynamic and biogeochemistry of the SMTZ in marine sediments.

## 2 Study area

The Barents Sea is an epicontinental sea marked by an average depth of 230 m and extending from the coasts of northern Norway and Russia to the south, to 80° N and bordered to the east by the coast of Novaya Zemlya and to the west by the Norwegian Sea. In the present study, we focus on sediment cores collected from Ingøydjupet, an NW-SE-stretching ice-carved trough located in

the southwestern sector of the Barents Sea (Fig. 1a). Ingøydjupet area has water depth less than 500 m and seafloor morphology includes numerous iceberg plough-marks, mega-scale lineations as well as glacial moraines, indicating a dynamic history of ice sheet advance and retreat since the Pliocene (Winsborrow et al., 2010). This area is associated with widespread subsurface fluid-flow systems and major Mesozoic hydrocarbon reservoirs (Chand et al., 2009; Vadakkepuliyambatta et al., 2013). Core locations are spread over an area of ~ 3 km² in the NE part of Ingøydjupet (Fig. 1b), about 10 km from a gas field (Caurus field; well

no.7122/2-1 in Fig. 1b) (Norwegian Petroleum Directorate, factpage.npd.no), at water depths ranging from 370 m to 410 m.



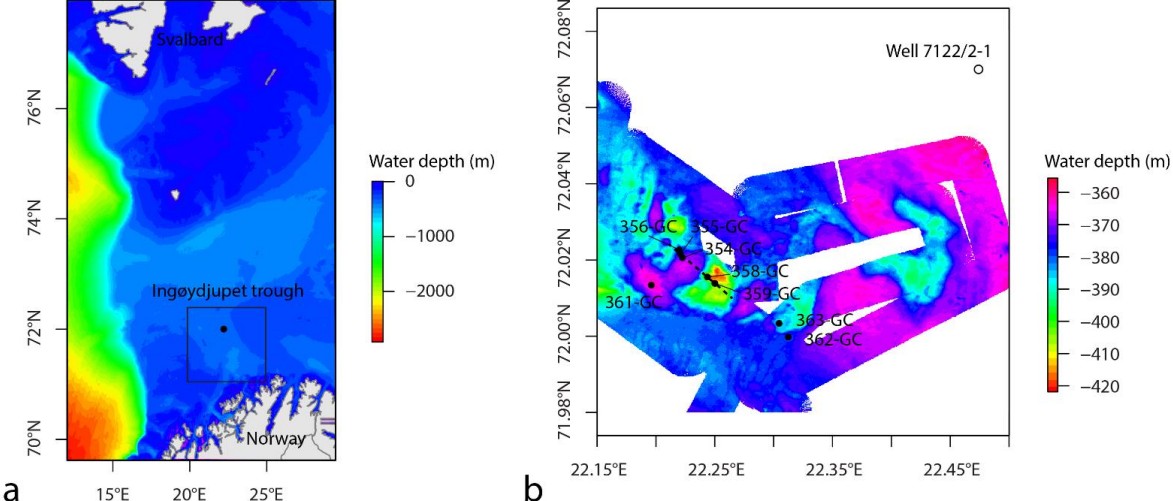

**Figure 1. Bathymetry map of the Barents Sea and of the studied area with indication of gravity core locations. a) Location of the study area (filled circle) in Ingøydjupet trough (box), SW Barents Sea. Bathymetry data from Gebco.net. b) Locations of gravity cores sampled for pore water and gas and discussed in this study (filled circles). The location of well no. 7122/2-1 associated with the discovery of the Caurus gas field in 2008 (see factpage.npd.no). Bathymetry acquired during expedition CAGE20-3 (cage.uit.no/). Dashed line indicates the track of a seismic line discussed in the text.**

## 3 Materials and methods

### 3.1 Pore water and headspace gas sampling

We collected sediment pore waters and headspace gas samples during the expedition CAGE20-3 in 2020 onboard R/V Helmer Hanssen to Ingøydjupet trough, SW Barents Sea (Norway). For this study we focus on eight gravity cores (CAGE20-3-HH-354-GC, CAGE20-3-HH-355-GC, CAGE20-3-HH-356-GC, CAGE20-3-HH-358-GC, CAGE20-3-HH-359-GC, CAGE20-3-HH-361-GC, CAGE20-3-HH-362-GC and CAGE20-3-HH-363-GC, hereafter named 354-GC, 355-GC, 356-GC, 358-GC, 359-GC, 361-GC, 362-GC and 363-GC), whose distribution is reported in Fig. 1b. Plastic liners used to collect cores 354-GC, 355-GC and 356-GC were drilled prior to coring operations using 3 mm and 1,5 cm drill bits for pore water and bulk sediment sampling (for headspace gas analysis), respectively. We maintained a 10 cm sampling resolution, keeping 5 cm between the 3 mm holes and 1,5 cm holes. Immediately after core recovery, the cores were cut into 100 cm sections, and sampled for pore water and bulk sediment. Pore water samples were taken every 10-20 cm and extracted by pressure filtration through 0.2 µm cellulose acetate filters using 10 cm long rhizon samplers. Each pore water sample was split into 2 aliquots: 1) subsamples for DIC analyses (1 ml) were transferred to 1.5 ml micro tubes with screw caps and we added 10 µl of $HgCl_2$ to stop microbial activity. 2) Subsamples for sulfate analysis (> 1 ml) were transferred to 5 ml Eppendorf tubes and stored at -20°C.

We collected bulk sediment samples every 10-20 cm in cores 354-GC, 355-GC and 356-GC and from the bottom of each section for the other cores. A 5 mL syringe without the luer tip was used to collect 5 mL of sediments, that were transferred to a 20 mL serum vial containing two glass beads and to which we added 5 mL of 1M NaOH to stop microbial activity. The vial was immediately closed with a septum and an aluminum crimp seal and stored at 4°C for onshore analyses.





### 3.2 Sulfate analysis


Pore water sulfate ($SO_4^{2-}$) concentration was measured at the University of Bergen, Department of Earth Science (Bergen, Norway), via inductively coupled plasma - optical emission spectrometry using a Thermo Scientific iCAP7000. Repeated measurements of ERM CA016a certified material were used to check the accuracy and precision during the analyses: measured values agree within the uncertainty of the certified value.


### 3.3 Dissolved inorganic carbon (DIC) and gas analyses

Sample preparation for the measurement of concentration and carbon isotopic composition ($\delta^{13}C$) of dissolved inorganic carbon followed the protocol for marine pore waters described in Spötl (2005) and briefly reported here: 100 μl of 85% phosphoric acid ($H_3PO_4$) were added to 4.5mL vials and flushed with helium. 400 μl or 200 μl of sample was injected into the vial using a syringe and left to equilibrate for 24h at 25°C prior to analysis. Measurements of the isotopic composition of released $CO_2$ were done using


a Thermo Scientific MAT253 IRMS coupled to a Gasbench II at the Stable Isotope Laboratory at UiT - The Arctic University of Norway in Tromsø. Data are reported in‰ notation relative to the Vienna Pee Dee Belemnite (VPDB) standard and analytical error was better than 0.1‰ (1SD). DIC concentration was calculated according to peak heights from the IRSM results and by comparison with signal strength of a $NaHCO_{3(aq)}$ calibration curve. Headspace gas composition was measured on a ThermoScientific GC Trace 1310 gas chromatograph-mass spectrometer equipped with a Restek Rt-Alumina BOND/Na2SO4


column for hydrocarbon gas separation. The GC system was calibrated using 2 external standards at 100 ppm and one standard at 1000 ppm. Analyses were conducted at the the Stable Isotope Laboratory at UiT - The Arctic University of Norway in Tromsø. The isotopic composition of methane was measured in one sample collected at 305 cm in core 358-GC. Methane in the headspace gas was separated using a Restek MXT Molsieve 5Å gas chromatography column, ionized at 950°C and measured using a Nu Horizon IRMS. Calibration is done with in-house standard calibrated with the international USGS standards NGS-2 and USGS


NGS-3. Precision was better than 0.5‰.

### 3.4 Modelling

The pore water sulfate concentration profiles display a linear shape in all the examined cores, which indicates that sulfate reduction at these sites is at a steady state and in a diffusion-controlled system (Hensen et al., 2003; Kasten et al., 2003; Schulz, 2006). Diffusive sulfate fluxes (J, nmol cm$^{-2}$ d$^{-1}$) for each core were calculated from the sulfate concentration gradients using Fick's


First Law (Eq. 1):

$$J = \varphi \times Ds \times \frac{dC}{dz}$$ (1)

where φ is the sediment porosity, Ds the whole-sediment diffusion coefficient and $dC/dz$ is the concentration gradient (Boudreau, 1996). Porosity (φ) was determined onshore from the water loss per volume of sediment upon drying (Jørgensen and Parkes, 2010) in sediment samples collected from core 354-GC, and gave an average value φ = 0.64 ± 0.06 (SD; n = 3). Ds was obtained from


the free-solution molecular diffusion coefficients of sulfate, Do, corrected for tortuosity using the empirical formula of Iversen and Jørgensen (1993) (Eq. 2):

$$Ds = \frac{Do}{1+3\,(1-\varphi)}$$ (2)

where Do = 0.48 cm$^2$ d$^{-1}$ at 4 °C (Schulz, 2006). The concentration gradient $dC/dz$ was obtained by linear least squares fitting of the sulfate concentration profile (Borowski et al., 2000; Graves et al., 2017; Mazumdar et al., 2007; Schulz, 2006). Pore water


profiles were extrapolated to the depth of the SMTZ by linear regression (Graves et al., 2017). The diffusive sulfate flux into the



SMTZ feeds both oxidation of buried organic matter (OSR) and AOM, and also corresponds to total sulfate reduction rate (SRR$_{tot}$) (Jørgensen et al., 2019). In order to quantify AOM and OSR rates in the SMTZ, we calculated their relative contributions to the pore water DIC pool using a mixing model, based on mass balance equations Eq. (3) and Eq. (4):

$$\delta^{13}C_{added} = f_{OSR} \times \delta^{13}C_{SOC} + f_{AOM} \times \delta^{13}C_{AOM} \tag{3}$$

$$f_{OSR} + f_{AOM} = 1 \tag{4}$$

where $\delta^{13}C_{added}$ represents the isotopic composition of bulk DIC released into pore water by biogeochemical reactions (i.e. OSR, AOM) and added to the initial background pore water DIC (buried seawater DIC) (Borowski et al., 2000; Hong et al., 2013; Hu et al., 2017, 2010; Ussler and Paull, 2008; Wu et al., 2016). In these equations, $f_{OSR}$ and $f_{AOM}$ are the fractions of DIC inputs (%) from OSR and AOM, respectively, and $\delta^{13}C_{SOC}$ (SOC = sedimentary organic carbon) and $\delta^{13}C_{AOM}$ indicate their isotopic composition.

$\delta^{13}C_{added}$ was obtained by linear regression of DIC concentration against the product of DIC concentration by its isotopic composition (DIC × $\delta^{13}C_{DIC}$) (Hu et al., 2010; Wu et al., 2016); the regression slope corresponds to $\delta^{13}C_{added}$. In our model we assumed an average $\delta^{13}C_{SOC}$ = -21.8‰ (Knies and Martinez, 2009) and $\delta^{13}C_{AOM}$ = -63.1‰, the latter representing the composition of DIC produced by anaerobic oxidation of methane in the study area ($\delta^{13}C_{CH4}$ = -54.1‰, our study), and assuming ~9‰ fractionation during AOM (Alperin et al., 1988; Beulig et al., 2019; Whiticar, 1999). The uncertainty on $f_{OSR}$ and $f_{AOM}$ was

estimated by solving the equations using $\delta^{13}C_{SOC}$ of pure marine and pure terrestrial organic matter end-members (Knies and Martinez, 2009) and is less than ± 0.05. We applied this model to cores 358-GC and 359-GC as the collected DIC samples reach the depth of the SMTZ (or close to it) and allow accurate modelling.

## 4 Results

### 4.1 Sulfate data

Pore water sulfate concentration profiles in all the examined cores have a linear shape and allowed an accurate calculation of sulfate gradients by linear regression. Diffusive fluxes calculated using Fick's First Law range from 1.5 nmol cm$^{-2}$ d$^{-1}$ (core 354-GC) to 12.0 nmol cm$^{-2}$ d$^{-1}$ (core 358-GC) and correspond to the total sulfate reduction rates in the SMTZ (SRR$_{tot}$) (Fig. 2, Tab. 1). The depth of the SMTZ, determined by linear extrapolation of the pore water profile to the depth of complete sulfate depletion, is consistent with sulfate fluxes (the greater the flux, the shallower the SMTZ) and ranges from 3.5 m to 29.2 m (Tab. 1). In all cores

except for 358-GC, the depth of the SMTZ was deeper than the length of the core (Fig. 2). The log-log linear relationship between SRR$_{tot}$ (nmol cm$^{-2}$ d$^{-1}$) and depth of the SMTZ (m) is described by the power law function: SRR$_{tot}$ = 43.284 ×SMTZ$^{-0.998}$.

| Sediment core | SRR$_{tot}$ (nmol cm$^{-2}$ d$^{-1}$) | SMTZ (m) | $\delta^{13}C_{added}$ (‰) | $f_{OSR}$ (%) | $f_{AOM}$ (%) | OSR (nmol cm$^{-2}$ d$^{-1}$) | AOM (nmol cm$^{-2}$ d$^{-1}$) |
|---|---|---|---|---|---|---|---|
| 354-GC | 1.5 | 29.2 | - | - | - | - | - |
| 355-GC | 1.7 | 25.2 | - | - | - | - | - |
| 356-GC | 4.5 | 9.8 | - | - | - | - | - |
| 358-GC | 12.0 | 3.5 | -45.0 | 44 | 56 | 5.3 | 6.7 |
| 359-GC | 10.2 | 4.5 | -48.4 | 36 | 64 | 3.7 | 6.5 |
| 361-GC | 2.2 | 20.1 | - | - | - | - | - |
| 362-GC | 3.2 | 12.7 | - | - | - | - | - |
| 363-GC | 4.8 | 9.0 | - | - | - | - | - |

Table 1. Results from pore water modeling. Total sulfate reduction rate (SRR$_{tot}$) is equal to the diffusive sulfate flux into the SMTZ. The depth of the SMTZ was obtained by linear regression of the sulfate concentration profiles. $\delta^{13}C_{added}$ indicates the composition of

bulk DIC released into pore water by OSR and AOM, and $f_{OSR}$ and $f_{OSR}$ are their respective contributions to the pore water DIC pool, calculated using a mixing model.

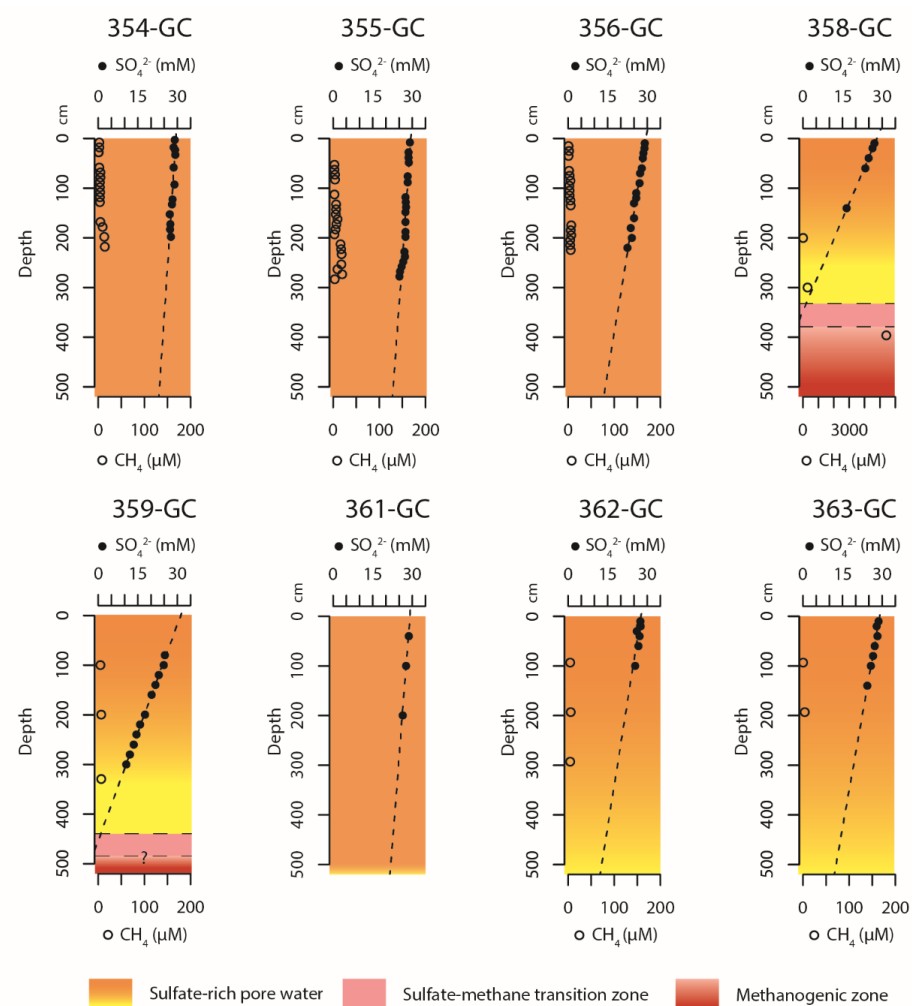

**Figure 2. Dissolved sulfate and methane concentration profiles in gravity cores collected during the expedition CAGE20-3 to Ingøydjupet trough, SW Barents Sea. The diagonal dashed line indicates the model fit to sulfate data using linear**

**regression.**

**4.2 Methane data**

        Methane concentration in the sediment cores was low (< 50 µM) except for 358-GC (Fig. 2), where the concentration value measured below the SMTZ reaches 5.4 mM. These observations validate our assumptions, i.e. the examined data reflect a diffusive-controlled system with complete methane consumption in the SMTZ by AOM. Hydrocarbon gas diffusing into the SMTZ

is dominated by methane, representing 99.8% of the total hydrocarbon gas fraction (core 358-GC). The carbon isotopic composition of the headspace gas sample collected at in core 358-GC is $\delta^{13}C_{CH4}$ = -54.1‰ VPDB.



### 4.3 DIC data

Dissolved inorganic carbon in the examined pore water samples ranges in concentration from 0.7 mM to 1.0 mM in core 355-GC, from 0.9 mM to 1.9 mM in 358-GC, from 0.7 mM to 2.7 mM in 359-GC and from 0.7 mM to 1.7 mM in 363-GC. The

carbon isotopic composition of DIC ($\delta^{13}C_{DIC}$) ranges from -12.1‰ to -2.4‰ in 355-GC, from -29.4‰ to -13.1‰ in 358-GC, from -36.8‰ to -9.2‰ in 359-GC and from -17.1‰ to -5.5‰ in 363-GC. The $\delta^{13}C_{added}$ values obtained from linear regression of DIC against DIC $\times$ $\delta^{13}C_{DIC}$, are -45.0‰ and -48.4‰ for 358-GC and 359-GC, respectively (Fig. 3, Tab. 1).

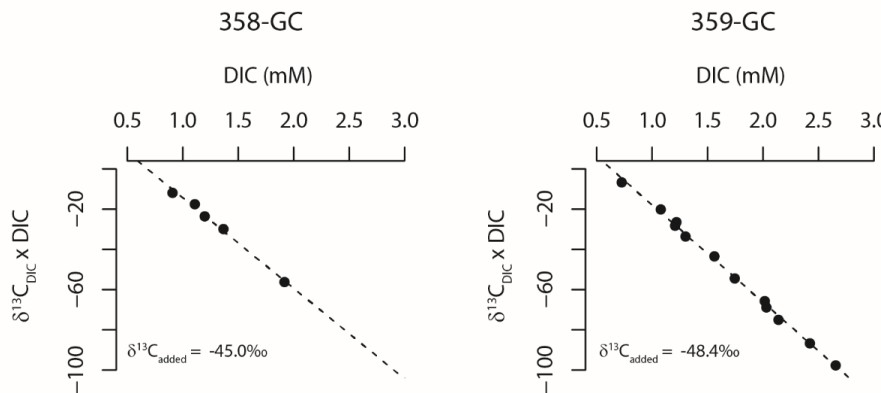

**Figure 3. Plots showing DIC concentration against the product of DIC by their $\delta^{13}C$ composition (DIC $\times$ $\delta^{13}C_{DIC}$), used to calculate the isotopic composition of bulk DIC added to pore water by OSR and AOM ($\delta^{13}C_{added}$).**

### 4.4 Mixing model results

The fraction of sulfate diffusing into the SMTZ, which feeds OSR ($f_{OSR}$) is 44% in 358-GC and 36% in 359-GC (Tab. 1). These values correspond to OSR rates of 5.3 nmol cm$^{-2}$ d$^{-1}$ and 3.7 nmol cm$^{-2}$ d$^{-1}$ (Tab. 1), respectively. Conversely, AOM in cores

358-GC and 359-GC contributes to the 56% and 64% ($f_{AOM}$) of total sulfate reduction in the SMTZ, corresponding to AOM rates of 6.7 nmol cm$^{-2}$ d$^{-1}$ and 6.5 nmol cm$^{-2}$ d$^{-1}$ (Tab. 1).

## 5 Discussion

### 5.1 Organoclastic sulfate reduction and anaerobic oxidation of methane at the SMTZ

The pore water concentration profiles provided important details about sulfate reduction in the sediment. Specifically, the

linear shape of the sulfate profiles found in all the cores is indicative of steady-state conditions and diffusive sulfate transport into the SMTZ (Kasten et al., 2003; Schulz, 2006) (Fig. 2). We quantified the sulfate flux from linear regression of the concentration data and extrapolated the profile to the depth of complete sulfate depletion, defining the position of the SMTZ within the sediments for each core site. The depth of the SMTZ found in cores 354-GC, 355-GC and 361-GC (>20 m below the seafloor) agrees well with previous observations from sediment cores collected from the NW Ingøydjupet trough (SMTZ located at 37 m; Nickel et al.,

2012). These SMTZs are deeper than the global average value for outer shelf (50-200 m; SMTZ depth = 4.0 ± 3.1 m) and slope (200-2000 m; SMTZ depth = 12.8 ± 12.1 m) settings (Egger et al., 2018), indicating reduced microbial activity due to the low sedimentary organic matter content in this area (TOC < 0.5%; Knies and Martinez, 2009). The methane concentration above the





extrapolated SMTZ depths is negligible (Fig. 2), indicating a complete methane consumption within the SMTZ by AOM microbial consortia. Considering a 1 : 1 ratio between sulfate and methane consumption by AOM in the SMTZ (Boetius et al., 2000), the

diffusive sulfate flux into the SMTZ would correspond to the upward diffusive methane flux, and $SRR_{tot}$ = AOM rate (Graves et al., 2017; Hoehler et al., 2000). However, recent studies (Beulig et al., 2019; Jørgensen et al., 2019), demonstrated that OSR may also contribute to the overall sulfate reduction within the SMTZ, resulting in flux discrepancies. Therefore, $SRR_{tot}$ can only provide the upper threshold value for AOM rates (Graves et al., 2017). In agreement with those studies, our results indicate that AOM contributes to 56-64% of total sulfate reduction at the SMTZ (Tab. 1). Conversely, OSR accounts for the 36-44% of total sulfate

reduction. These proportions are consistent with published range of values obtained in other marine settings worldwide marked by variable depth of the SMTZ ( < 20 m), e.g. Baltic Sea  (OSR = 14-59% $SRR_{tot}$)(Jørgensen et al., 2019), Ulleung Basin (OSR = 10-30% $SRR_{tot}$)(Hong et al., 2013), Santa Barbara Basin (OSR = 35-45% $SRR_{tot}$) (Komada et al., 2016), Gulf of Mexico (OSR up to 50% $SRR_{tot}$) (Hu et al., 2010). Our observations seem confirming that in different diffusive systems at steady-state, OSR contribution to total sulfate reduction is approximatively the same both in shallow (< 5 m) SMTZs (our study) and deeper SMTZs

(previous studies) as discussed by Egger et al. (2018).

The modeled reaction rates represent the net rates, resulting from all primary (e.g. AOM, OSR) and secondary reactions (e.g.re-oxidation processes), and gross rates are only determined by radiotracers techniques (Beulig et al., 2019; Jørgensen et al., 2019; Kasten et al., 2003). A recent study by Beulig et al. (2019) on cores from the Baltic Sea, showed that up to 60% of organic matter oxidation taking place within the SMTZ first produced methane, the latter becoming subsequently oxidized by AOM,

stimulating a cryptic methane cycle (Beulig et al., 2019). This process results in an additional release of [13]C-depleted DIC into pore water that is not linked to the diffusive methane flux from below the SMTZ, but from in-situ methanogenesis. In order to provide a rough estimation of the potential effect of this process on the overall results of our model, we recalculated OSR and AOM contributions to $SRR_{tot}$, by adding a third component in the mixing model, represented by AOM related to in-situ microbial methane production and hereafter named $AOM_2$. AOM related to the diffusive methane flux is hereafter named $AOM_1$. As

boundary conditions for the new model, we constrained $f_{AOM2}$ to be $\leq$ $f_{OSR}$, and $\delta^{13}C_{AOM2}$ = - 80‰ (as an example, we assumed a hypothetical microbial $\delta^{13}C_{CH4}$ ~ -70‰ corrected for AOM-induced fractionation). For the two $\delta^{13}C_{added}$ values obtained in our study, $\delta^{13}C_{added}$ = -45.0‰ (core 348-GC) and $\delta^{13}C_{added}$ = -48.4‰ (core 359-GC), we obtained $f_{AOM2}$ values of 0.25 and 0.29, respectively, corresponding to 25% and 29% $SRR_{tot}$. $f_{AOM1}$ values are 0.21 (core 358-GC) and 0.24 (core 359-GC), corresponding to 21% and 24% $SRR_{tot}$. We note that $f_{AOM}$ ($f_{AOM1}$ + $f_{AOM2}$) values, and consequently also $f_{OSR}$ values, are very close to the values

obtained from the two-component model, demonstrating the reliability of the results obtained using that approach (Tab. 1).

### 5.2 Factors influencing SMTZ dynamic in the study area

The SMTZ represents an important oxidation barrier for methane in marine sediments and limits methane emissions into the hydro- and atmosphere (Weber et al., 2019). Its depth below the seafloor is a valuable qualitative proxy for methane fluxes (e.g. Borowski et al., 1999, 1996). At a regional-scale, methane fluxes in non-hydrate-bearing sediments are mainly controlled by

rates of methanogenesis, which in turn is influenced by tectonic and glacial processes (loading and erosion) affecting sediment deposition and burial history of the area and kerogen evolution (Ferry and Lessner, 2008). The sediment cores in this study were collected within a 3 km[2] area so we rule out the possibility that the different methane gradients are due to differential in-situ methanogenesis. 3D seismic data shows a high-amplitude anomaly in the shallow subsurface of the study area (Fig. 4), providing strong evidence for gas accumulation in the sediments. In particular, the spatial distribution of this anomaly perfectly matches with

the location of cores 358-GC and 359-GC (Fig. 4), which are the cores associated with the shallowest SMTZ (< 5 m) in our study.





Therefore, we confidently interpret the variability in modeled sulfate reduction rates and depths of the SMTZ (Tab. 1) to reflect the differential distribution of gas accumulation in the subsurface and associated upward methane fluxes (Fig. 5).

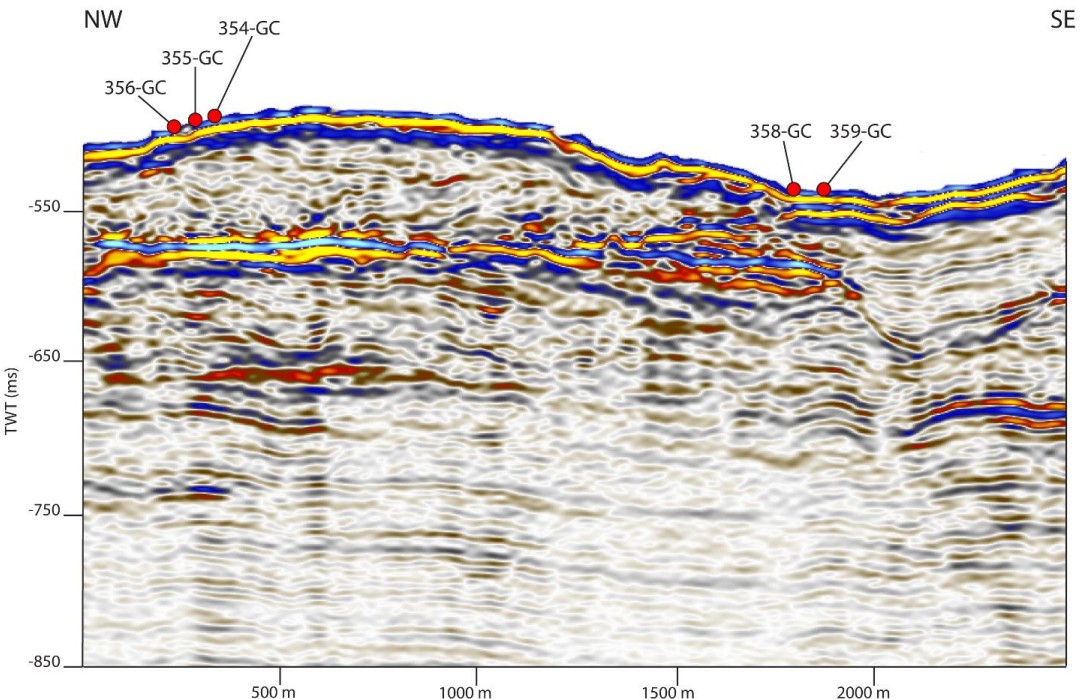

**Figure 4. Seismic line of Fig. 1b, showing a high-amplitude anomaly in the shallow subsurface due to gas accumulation. The spatial**
**distribution of the seismic anomaly matches with the location of gravity cores 358-GC and 359-GC, which are associated with the highest sulfate reduction rates and shallowest SMTZ. Interpretation was conducted using 3D seismic data available from the Norwegian Petroleum Directorate (npd.no).**

        The carbon isotopic composition of the gas sample collected at 305 cm from core 358-GC ($\delta^{13}C$ = -54.3‰) is similar to
the average isotopic composition of methane in Mesozoic successions from a nearby well (Well no. 7122-2-1, Caurus field, factpage.npd.no). We could not measure the hydrogen isotopic composition for our gas sample, due to limited sample size, and the determination of the gas source was out of the aim of this study. However, it is likely that the gas accumulated in the subsurface, and visible in seismic section, has a similar mixed microbial-thermogenic origin as the gas found in well no. 7122-2-1 located only ~ 10 km away. In conclusions, the data obtained in this study clearly highlight a link between subsurface methane gradients, sulfate
reduction rates and depth of the SMTZs (Fig. 5), providing new data from the SW Barents Sea that integrates previous global models and might help predict the fate of methane in marine sediments of continental margins affected by ocean warming.

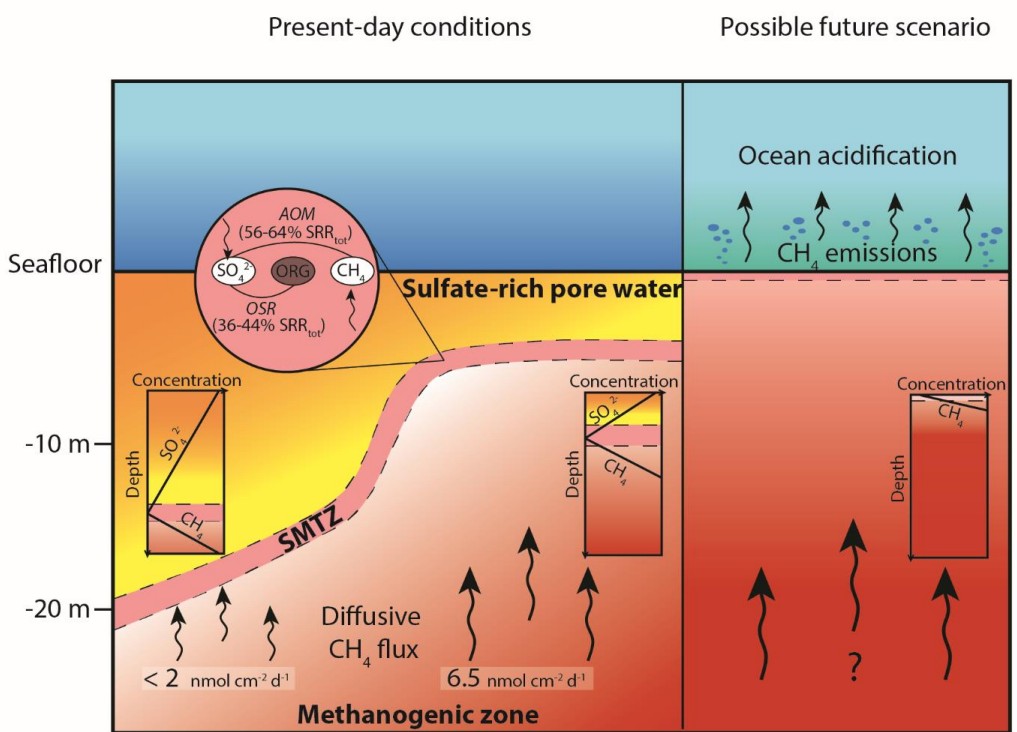

**Figure 5. Simplified model showing the relationship between diffusive methane fluxes and SMTZ depth in the investigated sediment cores in Ingøydjupet trough, SW Barents Sea. Within the SMTZ, OSR and AOM account respectively for 36-44% and 56-64% SRR$_{tot}$.**
**Ocean warming in the Arctic may lead to enhanced methane production and shoaling of the SMTZ, potentially leading to seafloor methane emissions with implications for ocean acidification. The same possible future scenario involves continental shelf areas where gas hydrate and submarine permafrost may undergo climate-driven destabilization. Box plots of sulfate and methane concentration profiles refers to steady-state conditions and starts from the seafloor; not in scale. AOM = anaerobic oxidation of methane; OSR = organoclastic sulfate reduction; ORG = organic matter; SMTZ = sulfate-methane transition zone.**

**6 Conclusions**

In this study we investigated the geochemical composition of pore water ($SO_4^{2-}$ and DIC concentration, $\delta^{13}C_{DIC}$) and gas ($CH_4$, $\delta^{13}C_{CH4}$) samples collected from 8 gravity cores from the Ingøydjupet trough, SW Barents Sea. Sulfate concentration profiles in all cores show a linear shape indicating steady-state conditions and a diffusion-controlled solute transport system. The SMTZ shows a wide range of depths below the seafloor, from 3.5 m to 29.2 m, and all methane is efficiently consumed by AOM within
the sediment. The diffusive sulfate fluxes into the SMTZ feeding organoclastic sulfate reduction and anaerobic oxidation of methane correspond to 1.5 nmol cm$^{-2}$ d$^{-1}$ and 12.0 nmol cm$^{-2}$ d$^{-1}$. Mixing modeling on two cores indicated that total sulfate reduction at the SMTZ is related for 36-44% to organoclastic sulfate reduction and for 56-64% to anaerobic oxidation of methane, corresponding to rates of 3.7-5.3 nmol cm$^{-2}$ d$^{-1}$ and 6.5-6.7 nmol cm$^{-2}$ d$^{-1}$, respectively. In the study area, the shallowest SMTZs are associated with a spatially-localized subsurface gas accumulation, clearly highlighting how small changes in sulfate reduction
rates in the order of ~ 10 nmol cm$^{-2}$ d$^{-1}$ linked to subsurface methane gradients result in large vertical shifts in SMTZ position (> 20 m). Due to the proximity (~ 10 km) of this area to a nearby gas field, and based on the isotopic composition of methane in the examined sample, we suggest that the gas diffusing toward the seafloor at the Ingøydjupet site has a mixed microbial-thermogenic source and migrated from Mesozoic reservoirs. The results of this study help better understand SMTZ dynamic in marine sediments



of continental margins and are of critical importance to predict the response of the microbial methane filter to future increase in
methane fluxes due to climate change.

**Data availability**

The datasets used for this study are openly available in Dataverse.NO at https://doi.org/10.18710/LI7QNQ.

**Author contributions**

C.A. and G.P. conceived the study. C.A. and K.A.W. conducted the offshore sampling operations. C.A.
wrote the initial manuscript and C.A. and K.A.W produced figures and tables. C.A., S.B. G.P. and K.A.W.
all contributed to the discussion and improvement of the paper.

**Competing interests**

The authors declare that they have no conflict of interest.

**Acknowledgments**

This work was supported by the Research Council of Norway through its Centre of Excellence funding scheme for CAGE Centre
for Arctic Gas Hydrate, Environment and Climate, project number 223259. We thank Aker BP for supporting Claudio Argentino
and Kate A. Waghorn. We would like to acknowledge the captain and crew onboard R/V *Helmer Hanssen* for the assistance during
the expedition CAGE20-3. We are grateful to Malin Wage (CAGE-UiT) for support during onboard sampling, and to Matteus
Lindgren (UiT) for the analysis of dissolved inorganic carbon and gas composition.

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
