# Peer review of "Sulfate reduction and anaerobic oxidation of methane in sediments of the South-Western Barents Sea"

_Biogeosciences, 2021_

## Referee Comment (RC2)

[referee-annotated manuscript omitted]

---

## Author Comment (AC1)

**DIC fluxes at the SMTZ and estimations of OSR and AOM rates**

In the manuscript, the values of $\delta^{13}C_{added}$ for cores 358-GC and 359-GC (values in Table 1 and Fig. 3 in the manuscript) indicate the composition of the diffusive DIC flux leaving the sulfate-methane transition zone (SMTZ), corresponding to $\delta^{13}C_{DIC,TOP}$ of equation Eq.1 (Wurgaft et al., 2019):

$$J_{DIC,OUT} \times \delta^{13}C_{DIC,OUT} = J_{DIC,TOP} \times \delta^{13}C_{DIC,TOP} - J_{DIC,BOT} \times \delta^{13}C_{DIC,BOT} \qquad \text{(Eq.1)}$$

where $J_{DIC,OUT}$ x $\delta^{13}C_{DIC,OUT}$ represents the net DIC production in the SMTZ after Eq.2 (Wurgaft et al., 2019); $J_{DIC,TOP}$ x $\delta^{13}C_{DIC,TOP}$ and $J_{DIC,BOT}$ x $\delta^{13}C_{DIC,BOT}$ represent the diffusive fluxes at the top and the bottom of the SMTZ (Fig. 1 below) that can be calculated using [DIC] × $\delta^{13}C_{DIC}$ vs. [DIC] mixing/net-reaction plots for pore water samples collected above and below the SMTZ (Beulig et al., 2019; Komada et al., 2016; Wurgaft et al., 2019), respectively.

$$J_{DIC,OUT} = J_{DIC,TOP} - J_{DIC,BOT} - J_{Ca,Mg} \qquad \text{(Eq.2)}$$

In Eq.2, $J_{Ca,Mg}$ is the flux of calcium and magnesium into the SMTZ, representing DIC removal by authigenic carbonate formation. The carbon fractionation associated with carbonate precipitation is negligible (Borowski et al., 2000; Komada et al., 2016; Wurgaft et al., 2019).

[Figure]

**Figure 1. Scheme showing the diffusive DIC fluxes entering and leaving the SMTZ and resulting in the net flux $J_{DIC,OUT}$ with composition $\delta^{13}C_{DIC,OUT}$.**

It is important to specify that $\delta^{13}C_{DIC,TOP}$ is equal to $\delta^{13}C_{DIC,OUT}$ only in the case of no authigenic carbonate precipitation ($J_{Ca,Mg}$ = 0) and negligible DIC fluxes from the methanogenic zone into the SMTZ ($J_{DIC,BOT}$ = 0). Based on these two assumptions, we consider $\delta^{13}C_{added}$ ($\delta^{13}C_{DIC,TOP}$ of Eq.1, 2) to reflect the bulk composition of DIC generated by OSR and AOM within the SMTZ (Case 1 in Table 1 below). Therefore, we can use $\delta^{13}C_{added}$ for calculating the relative contributions of organoclastic sulfate reduction (OSR) and anaerobic oxidation of methane (AOM) to DIC production based on Eq. 3 and Eq. 4 of the manuscript (Komada et al., 2016; Wurgaft et al., 2019). For each mole of $SO_4^{2-}$ consumed by OSR, 2 moles of DIC are produced (ox = 0, $C_{org}$ as $CH_2O$)(Komada et al., 2016), whereas sulfate-driven AOM has 1 : 1 stoichiometry between $SO_4^{2-}$ consumed and DIC produced. The rates of OSR and AOM for core 358-GC are 2.6 nmol cm$^{-2}$ d$^{-1}$ ($f_{OSR}$ (% $SRR_{tot}$) = 22) and 9.4 nmol cm$^{-2}$ d$^{-1}$ ($f_{AOM}$ (% $SRR_{tot}$)=78) (Case 1, Table 1 below), respectively. OSR and AOM rates for core 359-GC are 2.9 nmol cm$^{-2}$ d$^{-1}$ ($f_{OSR}$ (% $SRR_{tot}$) = 28) and 7.3 nmol cm$^{-2}$ d$^{-1}$ ($f_{AOM}$ (% $SRR_{tot}$)=72). Based on these considerations, the values reported here should replace the ones previously reported in the manuscript. We do not have pore water data for the deep sediment section below the SMTZs and therefore we cannot solve Eq. 1 and Eq. 2 for more accurate models. The effect of carbonate precipitation

and deep DIC fluxes into the SMTZ would change the fraction of AOM-related DIC production in the SMTZ ($f_{AOM}$ (% $DIC_{OUT}$) of Table 1 below). To better display this, we also modeled different cases assuming DIC removal by carbonate precipitation (Case 2), deep DIC flux ($\delta^{13}C_{DIC,BOT}$= -10‰) (Case 3), deep DIC flux ($\delta^{13}C_{DIC,BOT}$= -20‰) (Case 4) and deep DIC flux ($\delta^{13}C_{DIC,BOT}$= -30‰) (Case 5). $\delta^{13}C_{DIC,BOT}$ are depleted in $^{13}C$ compared to the source of methane (Whiticar, 1999). We assumed $J_{Ca,Mg}$~25% $J_{DIC,OUT}$ based on global estimates by Akam et al. (2020) and set $J_{DIC,BOT}$ =20% $J_{DIC,OUT}$ (lower than global average to fit the model). Results reported in Table 1 show that the AOM rates obtained for Case 1 ($J_{Ca,Mg}$ = 0; $J_{DIC,BOT}$ = 0) provide the minimum values that can explain the observed $\delta^{13}C_{added}$ when compared with the other cases. Conversely, the OSR rates obtained are the highest.

| | Case | 1 | 2 | 3 | 4 | 5 |
|---|---|---|---|---|---|---|
| | Assumptions | $J_{Ca,Mg} = 0$ $J_{DIC,BOT} = 0$ | $J_{Ca,Mg} \sim 25\% J_{DIC,OUT}$ $J_{DIC,BOT} = 0$ | $J_{Ca,Mg} = 0$ $J_{DIC,BOT} = 20\% J_{DIC,OUT}$ $\delta^{13}C_{DIC,OUT} = -10‰$ | $J_{Ca,Mg} = 0$ $J_{DIC,BOT} = 20\% J_{DIC,OUT}$ $\delta^{13}C_{DIC,OUT} = -20‰$ | $J_{Ca,Mg} = 0$ $J_{DIC,BOT} = 20\% J_{DIC,OUT}$ $\delta^{13}C_{DIC,OUT} = -30‰$ |
| 358-GC | $\delta^{13}C_{DIC,\,OUT}$ (‰) | -48.4 | -60.5 | -61.2 | -57.9 | -54.5 |
| | $f_{OSR}$ (% DIC) | 36 | 6 | 5 | 13 | 21 |
| | $f_{AOM}$ (% DIC) | 64 | 94 | 95 | 87 | 79 |
| | $f_{OSR}$ (% $SRR_{tot}$) | 22 | 3 | 2 | 7 | 12 |
| | $f_{AOM}$ (% $SRR_{tot}$) | 78 | 97 | 98 | 93 | 88 |
| 359-GC | $\delta^{13}C_{DIC,\,OUT}$ (‰) | -45.0 | -56.3 | -56.7 | -53.3 | -50.0 |
| | $f_{OSR}$ (% DIC) | 44 | 17 | 16 | 24 | 32 |
| | $f_{AOM}$ (% DIC) | 56 | 83 | 84 | 76 | 68 |
| | $f_{OSR}$ (% $SRR_{tot}$) | 28 | 9 | 8 | 13 | 19 |
| | $f_{AOM}$ (% $SRR_{tot}$) | 72 | 91 | 92 | 87 | 81 |

**Table 1. Results from DIC modeling and calculation of relative contributions of OSR and AOM to pore water DIC (f (% DIC)) and to total sulfate reduction within the SMTZ (f (% $SRR_{tot}$)) for different cases.**

**References**

Akam, S. A., Coffin, R. B., Abdulla, H. A. N. and Lyons, T. W.: Dissolved Inorganic Carbon Pump in Methane-Charged Shallow Marine Sediments: State of the Art and New Model Perspectives, Front. Mar. Sci., doi:10.3389/fmars.2020.00206, 2020.

Beulig, F., Røy, H., McGlynn, S. E. and Jørgensen, B. B.: Cryptic CH 4 cycling in the sulfate–methane transition of marine sediments apparently mediated by ANME-1 archaea, ISME J., 13(2), 250–262, doi:10.1038/s41396-018-0273-z, 2019.

Borowski, W. S., Hoehler, T. M., Alperin, M. J., Rodriguez, N. M. and Paull, C. K.: Significance of anaerobic methane oxidation in methane-rich sediments overlying the Blake Ridge gas hydrates, Proc. Ocean Drill. Progr. Sci. Results, 164(January), 87–99, doi:10.2973/odp.proc.sr.164.214.2000, 2000.

Komada, T., Burdige, D. J., Li, H.-L., Magen, C., Chanton, J. P. and Cada, A. K.: Organic matter cycling across the sulfate-methane transition zone of the Santa Barbara Basin, California Borderland, Geochim. Cosmochim. Acta, 176, 259–278, doi:10.1016/j.gca.2015.12.022, 2016.

Whiticar, M. J.: Carbon and hydrogen isotope systematics of bacterial formation and oxidation of methane, Chem. Geol., 161(1), 291–314, doi:10.1016/S0009-2541(99)00092-3, 1999.

Wurgaft, E., Findlay, A. J., Vigderovich, H., Herut, B. and Sivan, O.: Sulfate reduction rates in the sediments of the Mediterranean continental shelf inferred from combined dissolved inorganic carbon and total alkalinity profiles, Mar. Chem., 211(June 2018), 64–74, doi:10.1016/j.marchem.2019.03.004, 2019.